# Solvation-property relationship of lithium-sulphur battery electrolytes

Sang Cheol Kim [1,6], Xin Gao[1,6], Sheng-Lun Liao[2], Hance Su [1], Yuelang Chen [3], Wenbo Zhang [1], Louisa C. Greenburg[1], Jou-An Pan[1], Xueli Zheng [1], Yusheng Ye [1], Mun Sek Kim [2], Philaphon Sayavong [3], Aaron Brest[1], Jian Qin [2,7] ✉, Zhenan Bao [2,7] ✉ & Yi Cui [1,4,5,7] ✉

The Li-S battery is a promising next-generation battery chemistry that offers high energy density and low cost. The Li-S battery has a unique chemistry with intermediate sulphur species readily solvated in electrolytes, and understanding their implications is important from both practical and fundamental perspectives. In this study, we utilise the solvation free energy of electrolytes as a metric to formulate solvation-property relationships in various electrolytes and investigate their impact on the solvated lithium polysulphides. We find that solvation free energy influences Li-S battery voltage profile, lithium polysulphide solubility, Li-S battery cyclability and the Li metal anode; weaker solvation leads to lower 1st plateau voltage, higher 2nd plateau voltage, lower lithium polysulphide solubility, and superior cyclability of Li-S full cells and Li metal anodes. We believe that relationships delineated in this study can guide the design of high-performance electrolytes for Li-S batteries.

With the rapidly decreasing battery costs and growing concerns for climate change, electric vehicles are quickly becoming the future of the passenger vehicle market[1]. However, applications such as long-haul trucking and aviation remain difficult to electrify, and batteries with much higher energy densities are in demand[2]. Lithium-sulphur (Li-S) batteries are among the most promising candidates, as they have a theoretical specific energy exceeding 2500 Wh kg$^{-1}$ and >600 Wh kg$^{-1}$ batteries have been demonstrated[3].

The high energy density of Li-S batteries has roots in its multi-electron redox reaction, where sulphur assumes multiple oxidation states[3]. Intermediate sulphur species with oxidation states between elemental sulphur and lithium sulphide are lithium polysulphides (LiPS), which are readily soluble in many electrolyte solvents[4]. Solvated LiPS brings forth a unique aspect of the Li-S chemistry: active materials exist in both solid and solvated phases[4]. In various lithium-ion and lithium-metal battery chemistries with the active material confined to solid phase, full-cell reaction thermodynamics are independent of the electrolyte. However, because redox-active polysulphides are solvated in the electrolyte, the energetics of the polysulphide species and subsequently the full-cell thermodynamics depend on the electrolyte. In addition, the solubility of LiPS—a key factor in the Li-S battery performance as solvated LiPS can crossover to the anode and cause capacity degradation, electrolyte dry-out and self-discharge—will be heavily affected by the electrolyte[4]. These aspects amplify the importance of the electrolyte in Li-S batteries.

Solvation is a key determinant of the physicochemical properties of polysulphides and electrolytes. In efforts to tune solvation, chemistries such as ionic liquids[5], fluorinated ethers[6,7], high concentration electrolytes[8,9] and highly solvating electrolytes[10,11] have been developed. Strategies to tune the local solvation structures of Li+ have also been explored recently[12,13]. To understand the electrolyte solvation environment and how it affects Li-S battery performance, a host of spectroscopic, diffraction and imaging techniques have been employed[14–19]. In particular, recently developed nuclear magnetic

[1]Department of Materials Science and Engineering, Stanford University, Stanford, CA 94305, USA. [2]Department of Chemical Engineering, Stanford University, Stanford, CA 94305, USA. [3]Department of Chemistry, Stanford University, Stanford, CA 94305, USA. [4]Department of Energy Science and Engineering, Stanford University, Stanford, CA 94305, USA. [5]Stanford Institute for Materials and Energy Sciences, SLAC National Accelerator Laboratory, 2575 Sand Hill Road, Menlo Park, CA 94025, USA. [6]These authors contributed equally: Sang Cheol Kim, Xin Gao. [7]These authors jointly supervised this work: Jian Qin, Zhenan Bao, Yi Cui. ✉e-mail: jianq@stanford.edu; zbao@stanford.edu; yicui@stanford.edu

resonance (NMR) spectroscopy based techniques have unveiled important insights about the solvating power of various electrolytes, and how they affect Li-S battery performance[20,21]. Despite these important contributions, our understanding of solvation in Li-S battery electrolytes remains unsatisfactory, and quantitative descriptions of the relationship between solvation and Li-S battery performance are rare but needed.

In this study, we deploy the recently developed potentiometric measurement of the solvation free energy[22] to probe solvation-property relationships for Li-S battery electrolytes (Fig. 1a). The schematic summarises the four key properties of electrolytes that are correlated with solvation energy in this study. Solvation free energy is a fundamental parameter that governs the minimum work needed for solvation. We investigate a range of electrolytes with moderate to high solvation power and find that the voltage profiles depend on the solvation free energy: weak solvation leads to a lower 1st voltage plateau and a higher 2nd voltage plateau. LiPS solubility decreases with weaker solvation, and solvation free energy is found to be negatively correlated to the natural log of the dissociation constant ($K_{sp}$). LiPS greater solubility negatively affects the cycling Coulombic efficiency (CE) and the initial capacity. Lastly, solvation free energy is found to correlate to Li metal anode performance, morphology and interphase chemistry. We believe that these solvation-property relationships can serve as guidelines in designing high-performance electrolytes for Li-S batteries.

## Results

### Solvation free energy measurement of electrolytes

Measurement of the solvation free energy of Li⁺ is at the core of this study. In a recent study, we introduced a method to experimentally characterise the relative solvation free energy of electrolytes[22]. It employs potentiometry of an electrochemical cell with symmetric Li metal electrodes but asymmetric electrolytes in the two half cells. The half reactions consist of the Li/Li⁺ redox couple. Because the identical electrode terms cancel out, only the Li⁺ terms solvated in two different electrolytes remain, and the resulting net reaction is the transfer of Li⁺ between two electrolytes. The change in free energy in this process, which is the difference in the solvation free energies of Li⁺ in different electrolyte environments, gives rise to an electromotive force that can be measured. By holding one electrolyte as a reference electrolyte, it is

possible to characterise and compare the solvation free energy of various electrolytes relative to a reference electrolyte. It is important to note that while the standard notion for solvation free energy is defined for the dilute limit, in this work, we consider practically more relevant finite concentrations where the solvation free energy includes the concentration effects. In essence, our measured concentration-dependent solvation free energy is the chemical potential of dissolved ions.

In this study, we examine a series of ether-based electrolytes with moderate to high solvating power (Fig. 1b). Ethers are the most commonly used class of solvents for Li-S batteries, as other classes such as carbonates are unstable with the LiPS[23]. Adding fluorinated solvents with low solvating power or increasing concentration could tune the solvation energy, further balancing excessive side reactions and sluggish kinetics[4]. In addition, this range of electrolytes allows for an appropriate window to probe the thermodynamics; for example, we find that weakly (sparingly) solvating electrolytes tend to have overbearing kinetic limitations at room temperature (Supplementary Fig. 1).

We can identify a few trends in the measured Li⁺ solvation energies of different electrolytes. First, as salt concentration increases, we observe a more positive solvation energy that corresponds to weaker solvation. This is aligned with the Nernst equation, as the chemical potential of Li⁺ is higher at elevated concentrations. Second, mixtures of 1,2-dimethoxyethane (DME) and 1,1,2,2-tetrafluoroethyl-2,2,3,3-tetrafluoropropyl ether (TTE) solvents show that greater amounts of TTE leads to weaker (more positive) solvation free energies. This is in agreement with the fact that TTE acts as a diluent, rendering the electrolyte as a local high concentration electrolyte[24]. As the amount of TTE increases, the local concentration of Li⁺ increases, producing a more positive solvation energy. Third, the solvation strength of solvents increases in the order of 1,3-dioxolane (DOL)-DME-tetraethylene glycol dimethyl ether (G4), which is in agreement with previous reports in the literature[19,25,26].

### Solvation effect on Li-S battery voltage profile

The thermodynamic voltage is a core property of a battery. Battery voltage, alongside capacity, dictates the energy content of a battery cell. Battery voltage also impacts the compatibility and stability of other battery components. From a practical perspective, the

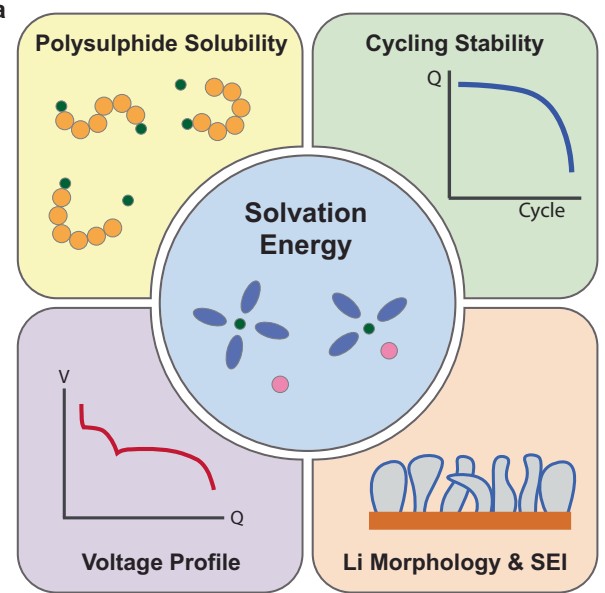

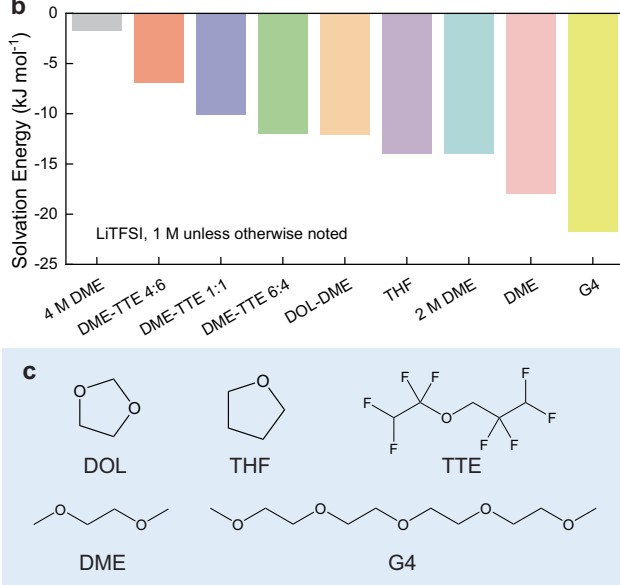

**Fig. 1 | Solvation free energy measurements of Li-S battery electrolytes. a** Schematic summarises the key electrolyte properties that are correlated with solvation free energy. **b** Measured solvation free energies of the electrolytes under investigation. **c** The molecular structures of the employed solvents.

thermodynamic voltage profile is an important indicator of the battery's state of charge during operation, which will indicate how much charge remains in the battery at any given moment. Although the thermodynamic voltage is unaffected by the electrolyte in most Li-based battery chemistries, the Li-S battery is different: it has a unique chemistry with active materials in both solid and solvated phases, allowing the electrolyte to play a role in the reaction thermodynamics. Indeed, different voltage profiles for different electrolytes have been observed[27,28], which can have profound implications on the energy density of the battery, but the mechanism remains unclear. We use the

measured solvation free energies to draw quantitative relationships to the Li-S battery voltage plateaus and propose a mechanism.

Figure 2a displays the discharge voltage profiles for two electrolytes: 1 M LiTFSI DME and 1 M LiTFSI DME-TTE (1:1 vol). The discharge c-rate is C/20, which makes the overpotential negligible and the profiles are essentially reflective of the thermodynamic properties. We can see that both electrolytes have two plateaus, with the first above 2.2 V and the second below 2.2 V. We observe that when TTE is added as a diluent, which weakens the solvation, the 1st plateau is lowered and the 2nd plateau is elevated. The trend holds for the series of ether-based

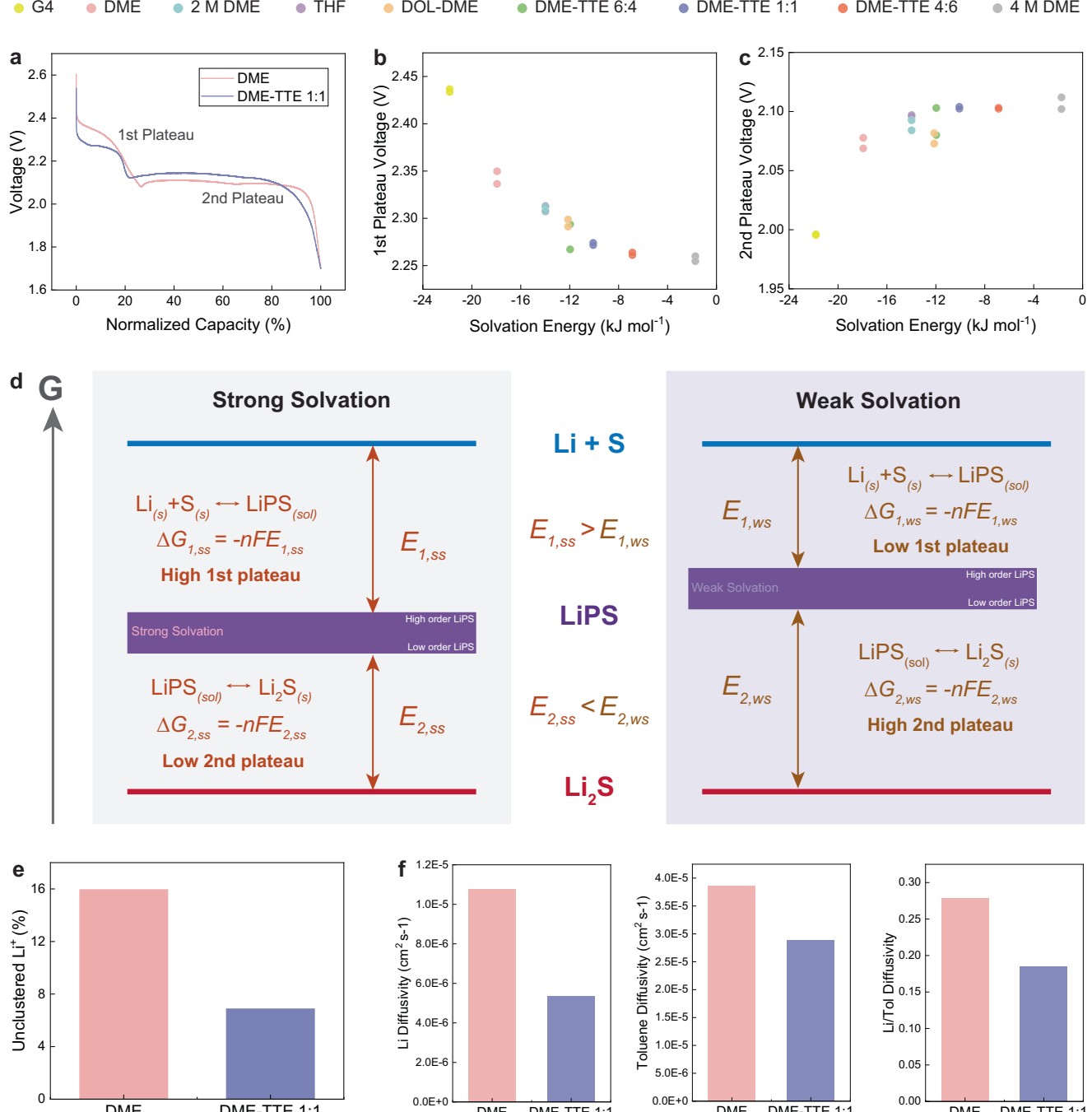

**Fig. 2 | Effect of solvation free energy on Li-S battery voltage profiles. a** Voltage profiles of 1 M LiTFSI DME and 1 M LiTFSI DME-TTE (1:1 vol) in Li-S batteries. **b** Correlation of 1st plateau voltage vs. solvation energy, showing decreasing voltage with weaker solvation. **c** Correlation of 2nd plateau voltage vs. solvation energy, showing increasing voltage with weaker solvation. **d** Proposed mechanism of the solvation free energy effect on Li-S battery voltage profile. **e** Molecular dynamics simulation of unclustered Li+ in the two electrolytes, showing that DME with stronger solvation has a larger fraction of unclustered Li+. **f** Diffusivity of Li and toluene and the ratios of the two parameters, showing that the less clustered DME has higher diffusivity.

electrolytes, where we see lower 1st plateau and higher 2nd plateau with weaker solvation (Fig. 2b, c, Supplementary Fig. 2), in agreement with previous observations[27–29]. This correlation can be extended to strongly solvating solvents such as dimethyl sulphoxide (DMSO) (Supplementary Fig. 3). The average voltages show a weak correlation with weaker solvation, which suggests that the voltage of the 2nd plateau, which has a larger capacity, bears greater weight on the average voltage (Supplementary Fig. 4).

In an effort to explain this correlation, the proposed mechanism is illustrated in Fig. 2d. The fully charged high energy state is composed of elemental Li and S, and the fully discharged state is $Li_2S$[23]. The free energies of the two end states are the same irrespective of the electrolyte as they are solid phases. The intermediate state is the solvated LiPS, whose free energy is deeply dependent on the solvation strength of the electrolyte. For the strong solvation case, the strong interaction with the electrolyte stabilises the LiPS and lowers its free energy, whereas the opposite case is true for the weak solvation case, where the free energy is higher. For both cases, the first plateau represents the reaction from elemental Li and S to solvated high order LiPS[23]. The low free energy of LiPS for the strong solvation case results in a large difference in free energies ($\Delta G_{1,ss}$). Because the free energy change upon this reaction is directly related to the negative of the cell voltage, $E_{1,ss}$, the 1st plateau of the strong solvation case has a relatively high voltage compared to the weak solvation case. The energetics of LiPS affect the second plateau as well, which represents the transformation of low order LiPS to $Li_2S$. For the strong solvation case with low free energy of LiPS, $\Delta G_{2,ss}$ and $E_{2,ss}$ are small, which leads to the low 2nd plateau voltage compared to the weak solvation case. In essence, solvation free energy of LiPS species dictates the change in free energy from Li+S to LiPS, and LiPS to $Li_2S$, which respectively determine the first and second plateau voltages.

These differences are intertwined with the solvation structure. We compared two electrolytes with different solvation strengths, 0.1 M $Li_2S_6$ in DME and DME-TTE 1:1, using molecular dynamics and diffusion-ordered spectroscopy-nuclear magnetic resonance (DOSY-NMR). (Supplementary Fig. 5–7). For the weakly solvating solvent formulation, DME-TTE 1:1 was used to ensure sufficient solubility of 0.1 M $Li_2S_6$ for the DOSY-NMR experiment. We find that DME, which provides strong solvation, is effective in dissociating $Li^+$ and has a relatively high percentage of unclustered $Li^+$ compared to DME-TTE (Fig. 2e). These results are further corroborated by Fig. 2f. As unclustered $Li^+$ content increases in strongly (highly) solvating DME, $Li^+$ self-diffusivity increases. Even when accounting for the viscosity differences that are reflected in toluene diffusivities, the Li/Tol diffusivity ratio is higher for DME case, further validating the differences in solvation structure that we find in MD simulations.

It is important to note that the proposed mechanism in Fig. 2d is a simplified model. It is well-established that LiPS is a series of $Li_2S_x$ compounds that are in equilibrium with each other[23], and the reaction pathway involves multiple LiPS lithiation reactions. Describing the precise reaction pathway and the involved LiPS species is an nontrivial task. However, we believe that our model circumvents this problem by describing the species as a single band of LiPS. This simplification is reasonable for explaining the shifts in voltage profiles, because all soluble LiPS species are likely to be affected in similar ways by different solvating environments; in weak solvation the entire band is shifted upwards in energy whereas the opposite is true for strong solvation. The premise is supported by the fact that solubilities of different LiPS species are higher in strongly solvating solvents[4]. The intricate relationship between solvation energy and solubility will be discussed in the next section.

## Polysulphide solubility and Li-S battery cyclability
The solubility of polysulphides is a key factor in Li-S battery performance, including cycle and calendar life, self-discharge, and internal resistance[4]. We prepared electrolytes saturated with $Li_2S_6$, as shown in Fig. 3a, where electrolytes with stronger solvation appear darker. We deployed UV-Vis spectroscopy to quantify the solubility of the LiPS (Fig. 3b), which confirmed that solubility decreases with weaker solvation—a finding extensively discussed in the literature[4]. As a note, tetraethylene glycol dimethyl ether (G4) was not included in this list because it led to significantly different polysulphide speciation and a peak in shorter wavelengths, which is known to be the $S_3^{\cdot-}$ radical anions that form in strongly solvating electrolytes[4,29,30].

Although different parameters such as donor number and dielectric constant have been considered as descriptors for polysulphide dissolution and its effect on battery properties, there has not been a single parameter that sufficiently describes the phenomenon. Because solvation free energy directly governs the solvation behaviour and is a descriptor in fundamental thermodynamic relationships, we believe it describes solvation and solubility more accurately. The solubility constant $K_{sp}$ is related to the free energy of dissolution $\Delta G_{diss}$ through Eq. (1)[31].

$$\Delta G_{diss} = -RT \ln(K_{sp}) \qquad (1)$$

$\Delta G_{diss}$ includes the negative of the lattice energy of $Li_2S_6$, which is a constant for all electrolytes, and the solvation energy terms of the individual species, leading to Eq. 2 that shows that solvation energy is proportional to the negative of natural log of the solubility constant. Although the solvation free energy term that we measure constitutes that of $Li^+$, here we assume that it scales with the solvation free energy of all species, meaning that a strong solvating solvent for $Li^+$ will be a strong solvating solvent for other ionic LiPS species. In addition, cations generally have smaller ionic radii and have larger solvation energies than anions and are likely to dominate the solvation energy of salts[32].

$$\Delta G_{solv}^{Li^+} \propto -\ln(K_{sp}) \qquad (2)$$

There may be several possible dissociation modes, as shown in Eqs. (3)–(5), and the equilibrium constant expression will vary depending on the reaction. Our data shows that the dissociation of $Li_2S_6$ most closely resembles Eqs. (3), (4), particularly the dissociation of $Li_2S_6$ into $Li^+$ and $LiS_6^-$ (Fig. 3c, Supplementary Fig. 8). Despite the excellent fit, it is important to note that $Li^+$ and $LiS_6^-$ will not be the only species in equilibrium. $Li_2S_6$, $S_6^{2-}$ and other speciation of polysulphides will also be present, although these species may not constitute the majority[30,33].

$$Li_2S_6(s) \leftrightarrow Li_2S_6(sol) \qquad (3)$$

$$Li_2S_6(s) \leftrightarrow Li^+(sol) + LiS_6^-(sol) \qquad (4)$$

$$Li_2S_6(s) \leftrightarrow 2Li^+(sol) + S_6^{2-}(sol) \qquad (5)$$

Polysulphide solubility is well known to have profound effects on battery performance[4,23]. We find that electrolytes with weak solvation and low LiPS solubility have high CE[20]. This is attributed to the shuttling effect to the metallic Li and the alleviation of this effect as LiPS solubility decreases[4,20]. We observe a similar trend with the initial discharge efficiency increasing with weaker solvation (Fig. 3e, Supplementary Fig. 9). This is somewhat surprising as polysulphide solubility is not expected to affect the amount of active material available. We believe that this is again related to the polysulphide shuttling effect; as polysulphide reacts with Li metal to form lithium-rich LiPS[23], active S is lost in the form of insoluble $Li_2S$ or $Li_2S_2$ on the anode or undergoes self-discharge, reducing the discharge capacity.

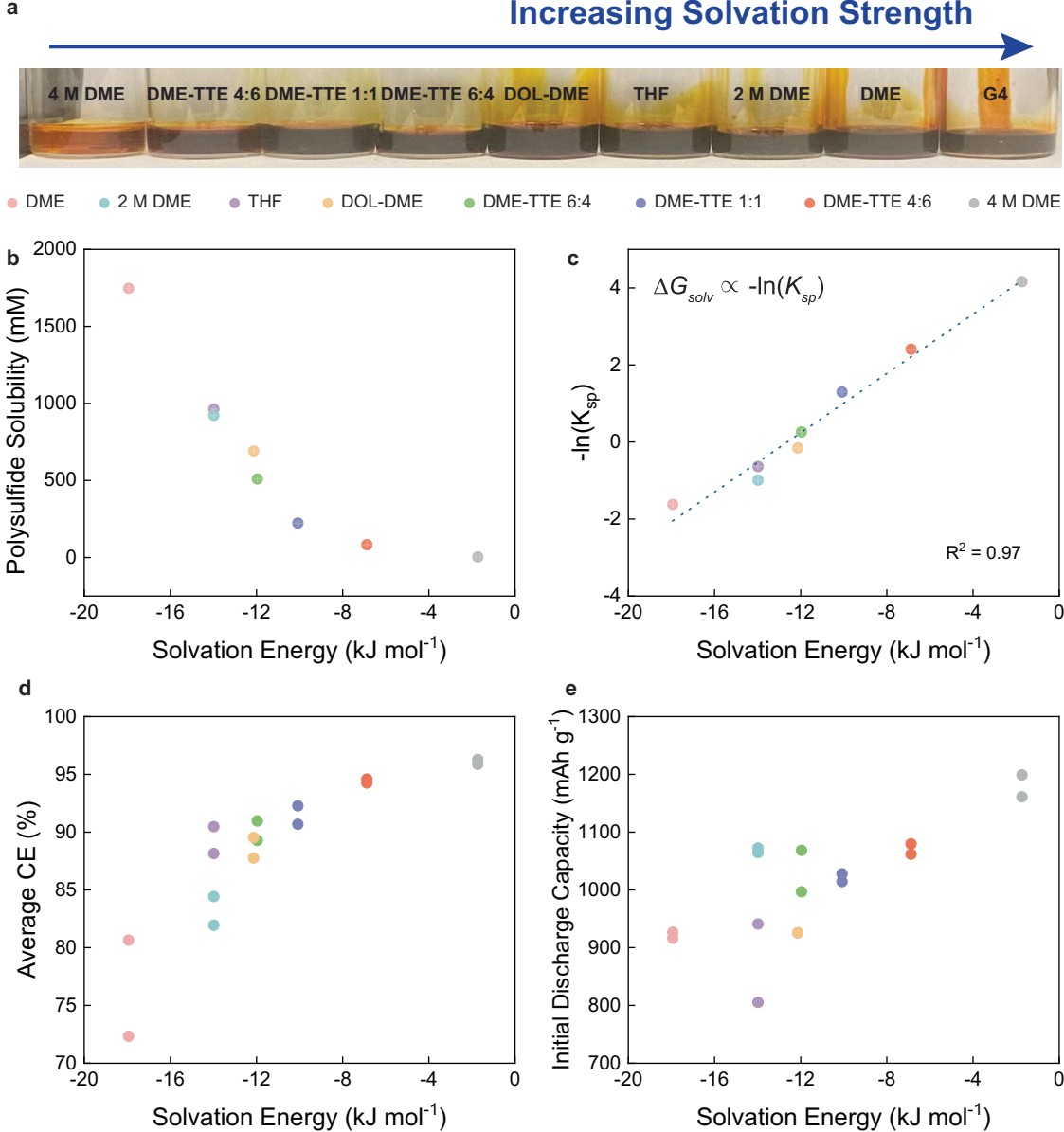

**Fig. 3 | Effect of solvation free energy on polysulfide solubility and Li-S battery cyclability. a** Digital photograph of electrolytes saturated with $Li_2S_6$. **b** Solubility of $Li_2S_6$ in the electrolytes quantified with UV-Vis spectroscopy. **c** Correlation of the negative of the natural log of the solubility constant with solvation energy.

**d, e** Correlations of Li-S battery CE averaged over 150 cycles and initial discharge capacity with solvation energy, showing that both metrics increase with weaker solvation.

Although weaker solvation is beneficial in improving CE and the initial discharge capacity, it can take a toll on the kinetics. Supplementary Fig. 10 shows that the 2$^{nd}$ plateau overpotential, obtained through the differences between cycling at 0.05C and 0.2C, increases with weaker solvation. This may be a result of insufficient concentration of LiPS that act as redox mediators to accelerate the charge-transfer reaction[13,34]. The kinetics seems to be play an important role in cycling stability as well, seen by the fact that capacity retention does not increase with weaker solvation, but rather peaks between solvation energies of −10 to −12 kJ mol$^{-1}$ (Supplementary Fig. 11), a phenomenon that corroborates with recent findings[35]. In addition, if solvation is further weakened and LiPS solubility is further suppressed, kinetic effects can take over where the discharge capacity decreases with weaker solvation (Supplementary Fig. 1)[4,23]. Therefore, regulating solvation strength is crucial for balancing electrochemical stability and kinetics to achieve optimal battery performance. With that, it is also important to note that solvation energy is not the sole factor in governing these properties; details in molecular structures and chemical reaction pathways that lead to different passivation layers are important contributors that must be taken into account for a holistic understanding.

## Lithium metal anode cyclability, morphology and interphase

When the anode has excess inventory of Li, as is the case for most Li-S batteries, the cell CE does not reflect that of the Li metal anode. However, the side reactions at the Li metal-electrolyte interface is undeniably important for the battery performance, as it leads to electrolyte dry-out, impedance build-up and the need for more excess Li and electrolyte that reduces the energy density[4]. We examined the Coulombic efficiencies of six electrolytes with and without $Li_2S_6$ using a modified Aurbach method (Supplementary Fig. 12). The Aurbach method commonly used to assess the Li metal CE cannot be used

directly when LiPS exists in the electrolyte. In a Li-Cu cell with an open circuit potential >2 V upon assembly, LiPS reduction reaction takes place prior to Li deposition, hindering accurate assessment of Li metal CE (Supplementary Fig. 13). We employed a modified protocol that replaces Cu with thin Li (20 μm) on Cu that circumvents the initial LiPS reduction reaction. 1 M LiTFSI G4, 1 M LiTFSI DME and 2 M LiTFSI DME, which possess relatively strong solvation and suffer instabilities at the Li metal anode, had CE results too low for the measurement and are not included in the plot (Fig. 4a).

Figure 4b shows that the CE of electrolytes without LiPS increases with weaker solvation, in agreement with previous observations (Supplementary Fig. 14)[22,36,37]. When the electrolyte is saturated with LiPS (Fig. 4c, Supplementary Fig. 15), which resembles the case of a Li-S battery with lean electrolyte (E/S ratio ≤ 5 mg μL$^{-1}$), a similar trend is observed except that the CE is overall lowered by about 3%. This result is surprising as LiPS is known to have a synergetic effect with LiNO$_3$ to stabilise the Li metal anode[38]. We conjecture that without LiNO$_3$, LiPS does not have stabilising effects and even has detrimental effects. This is supported by the scanning electron microscopy (SEM) images in Fig. 4d, e. It shows that 1 M LiTFSI DME without LiPS has areas of bare Cu, whereas the same electrolyte with LiPS has the entire active surface covered with Li metal. This suggests that the Li plated in 1 M LiTFSI DME has a smaller surface area, which corroborates the higher CE. Similar phenomena have been previously reported, where Li depositing in islands rather than covering the entire surface area has been correlated with improved CE[39,40]. A similar relationship can be observed for the 1 M LiTFSI DME-TTE 1:1 case, shown in Fig. 4f, g. A potential mechanism is that LiPS promotes Li nucleation over Li growth, which leads to the formation of small nuclei, increasing the overall surface area and side reactions. It is possible that an electrolyte decomposition layer on the current collector prior to Li deposition may lead to different nucleation behaviour. An alternative mechanism may be related to surface energy of Li in contact with the electrolyte. It has been suggested that the solvation energy of the electrolyte can dictate the Li surface energy at the Li-electrolyte interface, thereby impacting the morphology of Li[37]. It could be that the presence of LiPS alters the solvation energy that the high surface area Li morphology is favoured. Interphase chemistries were analysed with X-ray photoelectron spectroscopy (XPS) in Fig. 4h. We observe that the SEI is more inorganic-rich with higher fluorine and sulphur content when TTE is added to the electrolyte, which is consistent with literature reports[24]. When LiPS is incorporated into the electrolyte, the SEI becomes relatively richer in sulphur, which likely stems from the decomposition of LiPS.

## Discussion
In this study, we employed solvation free energy measured through potentiometric techniques as a metric to draw solvation-property relationships of Li-S battery electrolytes. We found that more positive solvation energy (weaker solvation) leads to lower 1st voltage plateau and higher 2nd voltage plateau, which stems from the free energies of LiPS solvated in electrolyte relative to the insoluble Li, S and Li$_2$S. Weaker solvation also leads to lower LiPS solubility and it was found that solvation free energy is directly proportional to -ln($K_{sp}$). Solubility is linked to both Li-S battery CE and initial discharge capacity. We also found that weaker solvation leads to superior Li metal CE and LiPS generally has a negative effect on the CE. Although weaker solvation has proven to be beneficial to full-cell and Li metal CE, weaker solvation is detrimental to the internal resistance of the cell.

Understanding these solvation-property relationships and trade-offs will be important for designing electrolytes for Li-S batteries. At the same time, despite these general trade-offs, some molecules and electrolyte chemistries may improve one performance metric without sacrificing others. Existing examples include LiNO$_3$ and DOL, which undergo unique molecular reactions at the Li metal interface and

significantly change the SEI properties. Finding additional novel electrolyte chemistries that can break out of the abovementioned trade-offs and concurrently achieve excellent electrochemical stability and kinetics will be an important future direction for Li-S battery electrolyte research.

## Methods
### Electrolyte preparation
Electrolytes were prepared in an Ar glovebox with O$_2$ concentration less than 0.2 ppm and H$_2$O concentration less than 0.01 ppm. All electrolyte materials were used as received after molecular sieving to remove trace amounts of water. LiTFSI (Solvay) was used as the salt DME (Sigma-Aldrich), G4 (Sigma-Aldrich), DOL (Sigma-Aldrich), THF (Sigma-Aldrich), TTE (SynQuest) were used as solvents. To test the solubility of LiPSs, Li$_2$S$_6$ was added into the different solvents to give a concertation of 2 M assuming it completely dissolves, and the emulsions were rested for 7 days to achieve equilibrium. To get the saturated Li$_2$S$_6$ solutions, undissolved solid was filtrated through a syringe filter with the pore size of 0.22 μm. In all electrolytes, LiNO$_3$ was not added as an additive.

### Electrochemical performance testing
S cathodes were fabricated using a slurry coating method. S-carbon composite (75 wt% S), was mixed with poly(vinylidene fluoride) (PVDF, Kynar HSV 900) binder, carbon black (Timcal Super-C65) in 84:8:8 mass ratio in N-methyl-2-pyrrolidone solution to form a slurry. The slurry was coated onto a carbon-coated aluminium foil by doctor blading and drying under vacuum at 60 °C for 48 h. The electrodes were cut into discs with mass loading of 1–2 mg cm$^{-2}$. The N/P ratio ranges are 29.9–59.8. Li-S cells (type 2032 coin cell) were assembled in an argon-filled glovebox using S cathodes (63 wt% S), Li metal anodes (~500 μm), separators (Celgard 2325), and the electrolytes (E/S ratio is 15 mg μL$^{-1}$). The Li-S cells were operated between 1.7 and 2.8 V using cyclers (Land Instruments) in an air-conditioned room without environmental chambers, at C/20 for the first two cycles, then at C/10 for three cycles, followed by continued cycling at C/5. Charge/discharge rates are calculated assuming the theoretical capacity of S (1675 mAh g$^{-1}$), therefore C/5 is 335 mA g$^{-1}$. Extraction of the 1st and 2nd plateau voltages were done by pinpointing the starting and the ending points of each plateau and finding the average voltage.

Li||Cu/thin Li half-cell cycling was tested by a modified Aurbach method as follows: (1) deposit 5 mAh cm$^{-2}$ Li on Cu/thin Li at 0.5 mA cm$^{-2}$ and strip to 1 V for formation cycle; (3) deposit 5 mAh cm$^{-2}$ Li on Cu at 0.5 mA cm$^{-2}$ as a Li reservoir; (4) repeatedly strip/deposit Li of 1 mAh cm$^{-2}$ at 0.5 mA cm$^{-2}$ for 10 cycles; (5) strip all Li to 1 V.

### Electrode and interphase characterisation
For electrode and interphase chatacterizations, Li-Cu coin cells were assembled using Celgard 2325 separators and different electrolytes and 1 mAh cm$^{-2}$ of Li was plated onto a Cu current collector at 0.5 mA cm$^{-2}$. Then the coin cell was disassembled in a glovebox and the electrode was extracted then washed in DME to remove any excess Li salt. The FEI Magellan 400 XHR was used for scanning electron micrographs. For the XPS results, a PHI VersaProbe 4 scanning XPS microprobe with an Al Kα source was used, where the SEI was characterised without sputtering.

### Electrolyte characterisation
Solvation-free energy measurements were done by using a potentiometric method, using a H-cell with Li metal as electrodes and asymmetric electrolytes. All results are referenced to 1 M LiFSI DEC as the reference electrolyte, and 3 M LiTFSI in DOL-DME was used as the salt bridge electrolyte. The open circuit voltage was measured for 1 min and recorded, and converted into solvation free energy using the equation $\Delta G = -nFE$, where $G$ is the Gibbs free energy, $n$ is the number

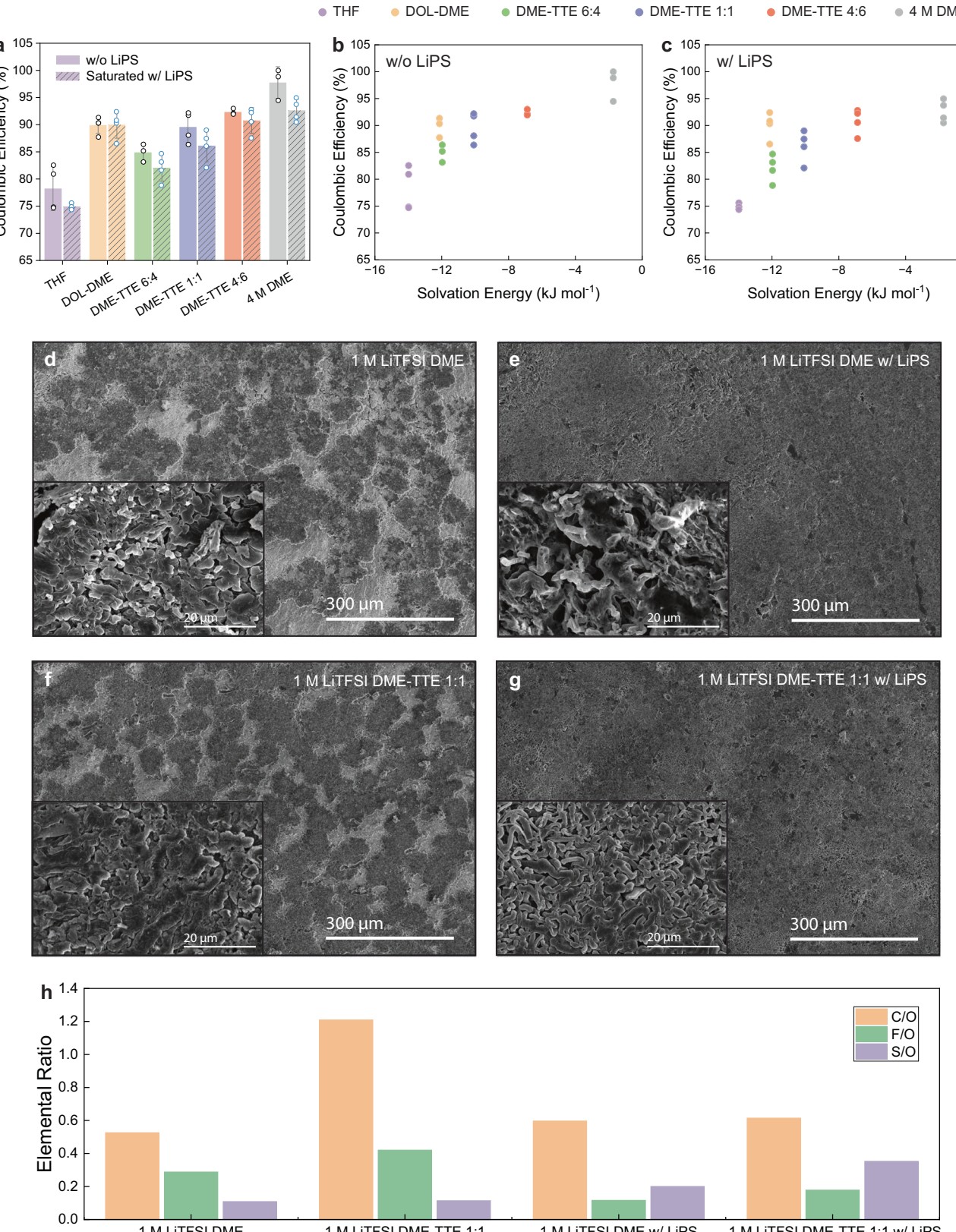

**Fig. 4 | Solvation effects on Li metal cyclability, morphology and interphase.**
**a** Li ‖ Cu CE of electrolytes with and without LiPS. **b, c** Correlations of Li ‖ Cu CE with solvation energy, showing increasing CE with weaker solvation, while the addition of LiPS leads to lower CE. **d–g** SEM micrographs of Li plated on Cu with 1 M LiTFSI DME and 1 M LiTFSI DME-TTE 1:1 with and without LiPS. **h** XPS elemental ratios of the SEI of the Li plated on Cu with 1 M LiTFSI DME and 1 M LiTFSI DME-TTE 1:1 with and without LiPS.

of electrons, $F$ is the Faraday constant and $E$ is the cell potential. UV-Vis spectroscopy was conducted using a Mettler Toledo Spectrophotometer UV7. For the DOSY-NMR experiments, a concentration of 0.1 M of $Li_2S_6$ was used in DME and DME-TTE 1:1 as electrolytes. Ten percent toluene was added as an internal reference. The NMR tube contained a co-axial tube with DMSO-d6. Varian 400 MHz spectrometer at 25 °C was used for all experiments. $^7Li$-pulsed field gradient measurements were performed to determine the diffusion coefficients using the standard dstebpgp3s pulse sequence. Array of gradient strength was set to 2.908–12.504 G/cm with 12 linear steps. High power 90° pulse (pw90) was 9 μs, acquisition time was 4 s, and recycling delay (d1) was 1 s. Gradient pulse duration (δ) was 9 ms. The Stejskal–Tanner equation was used to calculate the apparent diffusion coefficients.

## Molecular dynamics simulations

MD simulations were conducted using Gromacs 2021.3[41] with the optimised potentials for liquid simulations all atom (OPLS-AA) force field[42] for DME, and the reparametrised force field described in ref. 43. for the polysulphide $(S6)^{2-}$ ions and TTE. A concentration of 0.1 M of $Li_2S_6$ was used in DME and DME-TTE 1:1 as electrolytes.

The simulation box, subjected to three-dimensional periodic boundary conditions, was composed of 200 lithium ions $(Li^+)$, 100 polysulphide $(S6)^{2-}$, and an appropriate number of solvent molecules (DME and TTE) to match the prescribed concentration. The initial configuration contains randomly distributed ions and solvent molecules. Electrolytes and ions were equilibrated for 20 ns in a NPT ensemble using the Parrinello-Rahman barostat at 1 bar, with a time step of 1 fs, followed by 20 ns production run. MD trajectory data were saved every 1 ps. A Nosé-Hoover thermostat was applied throughout with a reference temperature of 300 K. The presented results were generated from the production run. The particle mesh Ewald method was used to calculate electrostatic interactions, with a real-space cut-off of 1 nm and a Fourier spacing of 0.16 nm. The Verlet cut-off scheme was used to generate pair lists. A cut-off of 1 nm was used for non-bonded Lennard-Jones interactions, and bonds with hydrogen atoms were constrained.

The visualisations were generated with VMD[43]. Solvation statistics were calculated using the MDAnalysis Python package[44,45].

## Data availability

All data is available in the main text or the supplementary information. Source data are provided with this paper.

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

## Acknowledgements
The battery and electrolyte measurement part were supported by the Assistant Secretary for Energy Efficiency and Renewable Energy, Office of Vehicle Technologies, of the U.S. Department of Energy under the Battery Materials Research (BMR) Programme and the Battery500 Consortium programme. Part of this work was performed at the Stanford Nano Shared Facilities (SNSF).

## Author contributions
S. C. K. and X. G. contributed equally. S. C. K., X. G. and Y. Cui conceived and designed the investigation. X. G., H. S. and A. B. conducted electrode fabrication and electrochemical performance testing. S. C. K. and J.-A. P. synthesised the electrolytes. S. C. K. conducted solvation free energy measurements. S.-L. L. conducted molecular dynamics simulations. Y. Chen conducted DOSY-NMR experiments. S. C. K., X. G. and H. S. conducted UV-Vis spectroscopy measurements. W. Z. conducted SEM characterisations. L. C. G. conducted XPS characterisation. P. S., M. S. K., X. Z., Y. Y. assisted with interpretation of results. J. Q., Z. B. and Y. Cui supervised the project. S. C. K., X. G. and Y. Cui co-wrote the paper. All authors discussed the results and commented on the manuscript.

## Competing interests
The authors declare no competing interests.
