## [Peer Review File · Nature Communications]

Solvation-Property Relationship of Lithium-Sulphur Battery ElectrolytesREVIEWER COMMENTS

Reviewer #1 (Remarks to the Author):

The authors determine solvation free energy measured through potentiometric techniques to better understand LiS batteries. They find that more positive solvation energy (weaker solvation) leads to lower 1st voltage plateau and higher 2nd voltage plateau, which stems from the free energies of LiPS solvated in electrolyte relative to the insoluble Li, S and Li₂S.

This work is fundamentally sound but I am not sure it is sufficiently novel to warrant publication in Nature Comms. There have been many similar studies and the lack of molecular-level understanding (i.e. from simulations) make this study perhaps less compelling that it could be. I suggest this paper is more appropriate for a more specialized journal.

Reviewer #2 (Remarks to the Author):

In this manuscript, the authors utilized solvation free energy as a metric to draw solvation-property relationships of Li-S battery electrolytes. They found that solvation free energy influences battery voltage profile, LiPS solubility, battery cyclability and the Li metal anode. The experiment and the logic were well organized, and the relationships delineated in this study could lead to exciting developments in the field and beyond. I think this manuscript can be accepted after the authors made the following major revisions.

1. Common sense suggests that covering the entire area would mean more uniform deposition than exposing the base, but in the paper: "We conjecture that without LiNO₃, LiPS it does not have stabilizing effects and even has detrimental effects. This is supported by the scanning electron microscopy (SEM) images in Fig. 4d-e. It shows that 1 M LiTFSI DME without LiPS has areas of bare Cu, whereas the same electrolyte with LiPS has the entire active surface covered with Li metal." Please explain this contradiction.
2. More details on the methods about how to obtain solvation energy should be offered for better understanding.
3. Does the level of the 2nd platform affect the capacity of the battery?
4. Why choose DME and DME-TTE 1:1 for comparison, instead of DME and 4DME, which can exclude the influence of TTE?
5. In Fig 3e, there seems to be no pattern between solvent energy and initial capacity, why?

Reviewer #3 (Remarks to the Author):

This manuscript (Solvation-Property Relationship of Lithium-Sulfur Battery Electrolytes) uses solvation free energy, measured by potentiometric techniques, as a metric to draw solvation-property relationships of Li-S battery electrolytes. Findings indicate that solvation free energy plays a crucial role in determining the voltage profile of Li-S batteries, as well as the solubility of lithium polysulfides (LiPS), cyclability of Li-S full cell, and the performance of Li metal anode. Weaker solvation free energy results in a lower first plateau voltage, higher second plateau voltage, lower LiPS solubility, and superior performance of the Li-S full cell and Lithium metal anode. But, the novelty is not enough for Nature Communications, and some expressions in the manuscripts are still inadequate or ambiguous. I suggest that it is not suitable for publication on this journal:

1. It is interesting that the initial discharging capacity rises with weak solvating electrolytes. The shuttling effect may play a significant role in increasing the internal resistance of the battery. Besides, the authors claim that the loss of sulfur in the anode side will decrease the initial discharging capacity, which also makes sense. However, a quantification of the effect on resistance and the loss of sulfur in the anode side is lacking to represent a more specific result.

2. In the discussion of voltage profiles, the authors represent that electrolytes may display different voltage plateaus and perform a theoretical simulation to explain this. There is no explanation of how voltage plateaus affect battery performance, which can be confusing.

3. Since the solvation energy is of great significance to the kinetics and thermodynamics of Li-S batteries, can authors provide a solvation energy standard (or value range) for the future optimization of Li-S electrolytes?

4. Will the solvation energy be the only factor that influences the CE and initial capacity? Adding fluorinated solvents into the electrolytes can also enhance the CE and lower the battery resistance. For instance, the generation of LiF by the decomposition of F-contained solvents can protect the anode from further corrosion of LiPSs. Discussion of this section is lacking in the manuscript.

5. The long-lasting effect of solvation energy on the full cell should be discussed. Authors should provide the long cycle performance of Li-S batteries using different electrolytes.

Reviewer #4 (Remarks to the Author):

Li-S batteries have the potential to replace lithium-ion batteries once the major technical challenges arising from polysulfide transport and the redox mechanism are understood. However, unlike the conventional transition metal oxide chemistry in Li-ion batteries, where the active materials remain solid throughout the redox process, the dissolution of lithium polysulfide species (the intermediate redox species of the S/S^{2-} redox couple) into the electrolyte poses a significant challenge. This is especially true when the average cell potential at different processes and cycling performance heavily depends on the solvation structure of lithium polysulfide in the electrolyte. Understanding this solvation-property-performance relationship is a challenging and crucial problem that requires careful attention.

The work presented in manuscript NCOMMS-23-26016 utilizes a relatively simple potentiometric measurement method that the authors recently reported. It aims to draw a correlation between the solvation free energy of the electrolyte and its impact on the solvated lithium polysulfide. The concept and experimental method presented in the manuscript are novel and should be reproducible. I agree that the work is significant enough to further advance Li-S technology. However, there are several specific questions I would like the authors to address (see below). In conclusion, I believe the work is worthy of publication in Nat. Commun. once the following comments have been adequately addressed.

1. The author's statement regarding the solvating energy and its impact on lithium polysulfide is based on a limited number of electrolyte systems, most of which are considered moderately solvating. However, it is well documented by literature review that polysulfide speciation and chemistry can be significantly altered by different classes of electrolyte systems (Joule, 2021, 5, 2323). Considering the importance of understanding the effects of different electrolyte systems, it would be valuable for the authors to comment on the potential impact of alternative solvents, including dimethylsulfoxide (DMSO) and

dimethylacetamide (DMA), which are classic examples. By studying the effects of such systems, the authors could provide a more comprehensive understanding of how different solvents influence the solvation energy of lithium polysulfides. This would help avoid overgeneralizing the concept and provide a more nuanced perspective on the solvation-property-performance relationship in Li-S batteries.

2. In the main text, the author acknowledges that most Li-S batteries often employ an excessive amount of Li anode, which poses challenges in accurately assessing the anode's health through Coulombic efficiency (CE). This acknowledgment is indeed correct. Therefore, it would be highly beneficial for the authors to provide the N/P ratio. While the Li metal thickness is provided in the experimental method, without knowing the area of the Li anode, it renders great difficulty for readers to assess the practical relevance and potential implications of the authors' findings in Fig. 4

3. In the introduction, the authors claim that Li-S batteries are the potential candidates battery technology for eVTOL applications. However, it is worth noting that eVTOL technology requires high volumetric density batteries that can deliver high power (i.e., high C-rate) during take-off and landing. Unfortunately, these are precisely the areas where Li-S battery technology currently faces challenges and limitations. To address this concern, it would be beneficial for the author to provide a more detailed comment explaining why they believe Li-S battery technology is still considered adventurous despite its current weaknesses in high-power applications. They could discuss potential strategies, advancements, or ongoing research efforts aimed at improving the power capabilities of Li-S batteries for demanding applications like eVTOL. This would provide a clearer understanding of the potential viability of Li-S batteries in the context of eVTOL technology.

4. In the experimental method, can the author clarify was LiNO_3 used as electrolyte additive?

5. I understand the authors colour-coded their figure (example include Fig. S3, 5, and 6.) that is consistent with the types of electrolytes used (first mentioned in Fig. 2). However, re-labeling the legend in these figure can provide clarity and avoid potential confusion for reader.

Response Letter to Reviewers for “Solvation-Property Relationship of Lithium-Sulfur Battery Electrolytes”

We are grateful to the reviewers for providing valuable comments and feedback to improve our manuscript. We believe that the comments and questions raised by the reviewers have helped us clarify our arguments and show the novelty of our work. In the letter below, we have responded to each question raised by the reviewers in blue. The revisions made to the main text of our manuscript have been highlighted to aid the reviewers.

Reviewer #1 (Remarks to the Author)

The authors determine solvation free energy measured through potentiometric techniques to better understand Li-S batteries. They find that more positive solvation energy (weaker solvation) leads to lower 1st voltage plateau and higher 2nd voltage plateau, which stems from the free energies of LiPS solvated in electrolyte relative to the insoluble Li, S and Li₂S.

This work is fundamentally sound but I am not sure it is sufficiently novel to warrant publication in Nature Comms. There have been many similar studies and the lack of molecular-level understanding (i.e. from simulations) make this study perhaps less compelling that it could be. I suggest this paper is more appropriate for a more specialized journal.

We would like to express our gratitude to the reviewer for reading and evaluating our manuscript. We are also grateful for the appreciation of the robustness of our work and how it can contribute to better understanding Li-S battery electrolytes. We also understand your concerns regarding the novelty of our work and the lack of molecular-level understanding, and we believe that we have inadequately communicated these points and have potentially caused confusion. We believe that this work provides critical quantitative insights regarding Li-S battery electrolytes that were previously unavailable. The experimental measurement of solvation free energy for Li battery electrolytes, which is the key enabler of this study, was developed recently by our group and was previously unavailable. The novel results obtained through our technique provide quantitative and convincing evidence to elucidate key solvation-property relationships for Li-S battery electrolytes. First, we find a striking correlation between Li-S battery voltage profile and solvation free energy. It is the first time that clear correlations between plateau voltages and solvation energy have been established, which was enabled by our quantitative characterization of the solvation energy. Moreover, we provide a mechanism based on thermodynamic principles that can explain the phenomena. Secondly, we establish a fundamental relationship between solvation free energy and polysulfide solubility. Indeed, it has been known that strongly solvating electrolytes have increased solubility of polysulfide species¹, but in this study, we extend the horizon of knowledge by establishing quantitative relationships between solvation free energy and solubility and also deducing the solubility constant (K_{sp}). Deducing these fundamental relationships and parameters are critical to better understanding the polysulfide dissolution problem, as well as to designing high-performance electrolytes. Lastly, we reveal the relationship between Coulombic efficiency (CE) of Li metal anodes and solvation free energy of Li-S battery electrolytes, where we find that weaker solvation leads to superior CEs. Overall, we believe that significant and novel quantitative solvation-property relationships are presented in this work, powered by our potentiometric method to characterize solvation free energy.

Regarding your comment on molecular-level understanding and simulations, we had already conducted molecular dynamics simulations, but we believe we had not made it as visible as it should be. In Fig. 2e, the clustering behaviors of 0.1 M Li_2S_6 in DME and DME-TTE 1:1 were simulated and summarized. Through these results we could see that the two electrolytes had different solvation behaviors and cluster formation properties, which arose from the differences in solvation strengths. Also, in Supplementary Fig. 4, the simulated boxes of 0.1 M Li_2S_6 in DME and DME-TTE 1:1 were displayed. Although we had provided these results, we agree with the reviewer that additional molecular-level understanding from simulations could be useful to augment this study. We have added Figs. R1-2 below as supplementary figures. Figs R1-2 show the radial distribution functions (RDF) and cumulative distribution functions (CDF) of Li-O coordination between Li^+ and solvents of DME and TTE. It is possible to see that the RDF and CDF of the two electrolytes do not differ appreciably. It means that the local solvation structures of the two electrolytes are not significantly different. However, the clustering behavior of the LiPS, as shown in Fig. 2e, is different as the unclustered Li^+ percentage is higher for the DME solvent. This result is corroborated by the diffusivity measurements shown in Fig. 2f.

Fig. R1. Radial distribution functions (RDFs) of Li^+ in 0.1 M Li_2S_6 in DME and 0.1 M Li_2S_6 in DME-TTE 1:1.

Fig. R2. Cumulative distribution functions (CDFs) of Li^+ in 0.1 M Li_2S_6 in DME and 0.1 M Li_2S_6 in DME-TTE 1:1.

Changes to the manuscript:

- Addition of Supplementary Figs. 5-6

Reviewer #2 (Remarks to the Author):

In this manuscript, the authors utilized solvation free energy as a metric to draw solvation-property relationships of Li-S battery electrolytes. They found that solvation free energy influences battery voltage profile, LiPS solubility, battery cyclability and the Li metal anode. The experiment and the logic were well organized, and the relationships delineated in this study could lead to exciting developments in the field and beyond. I think this manuscript can be accepted after the authors made the following major revisions.

We are grateful for the reviewer's appreciation of our work, its scientific rigor and its impact to the field. We are also thankful for the valuable and insightful comments provided below to improve our manuscript. We have revised our manuscript and provide detailed point-by-point response to your questions and comments below.

1. Common sense suggests that covering the entire area would mean more uniform deposition than exposing the base, but in the paper: "We conjecture that without LiNO₃, LiPS it does not have stabilizing effects and even has detrimental effects. This is supported by the scanning electron microscopy (SEM) images in Fig. 4d-e. It shows that 1 M LiTFSI DME without LiPS has areas of bare Cu, whereas the same electrolyte with LiPS has the entire active surface covered with Li metal." Please explain this contradiction.

Thank you for the great question. We believe that for a rather small capacity of 1 mAh/cm², Li metal surface area can be smaller when Li is deposited as islands rather than uniformly covering the entire current collector surface. As a thought experiment, we could imagine two scenarios: 1) flat Li deposition covering the entire current collector surface; 2) hemispherical Li deposition without covering the entire surface. For both cases, the deposited capacity is 1 mAh/cm². For the first case, the surface area will be 1 cm². With the second case, surface area will depend on the number of hemispheres. With the extreme scenario of 1 hemisphere, the surface area will be 0.025 cm², 1/40 of the surface area in scenario 1. Only when we have more than 100,000 hemispheres, with hemisphere radius of 13.63 μm, will the surface area of scenario 2 become larger than scenario 1. SEM images in Fig. 4 indicate that islands have radii of about 50 μm, and this would lead to only about a quarter of the surface area in the case of uniform coverage. Of course, this exercise is a simple and approximated thought experiment, but we believe it showcases that island-like morphologies can have smaller surface areas than uniform growth for small deposition capacities.

Table R1. Simulated surface areas of hemispherical Li deposition

# hemispheres	Hemisphere radius (μm)	Surface area (cm ²)
1	630	0.025
10	293	0.0542
100	136.33	0.117
1000	63.28	0.252
10000	29.37	0.542
100000	13.63	1.17
1000000	6.33	2.52
10000000	2.94	5.42

As for the performance benefits of this kind of deposition morphology, our group reported that when surface chemistry of current collectors was modified using atomic layer deposition, it was seen that Li existed as islands instead of wetting the entire current collector surface (Fig. R3).² It was also seen that the cycling performance improved with the Al₂O₂ surface layer. Similar phenomenon was also observed in another

work from our group, which compared the Li-Cu Coulombic efficiencies of two electrolytes (Fig. R4)³. 1 M LiFSI MTBE-toluene electrolyte that had island-like Li morphologies had high Coulombic efficiency compared to 1 M LiFSI DME that had complete coverage of Li.

Fig. R3. Li deposition morphology differences on control Cu current collector compared with Al₂O₃-modified current collector, adapted with permission from Oyakhire et al.²

Fig. R4. Li deposition morphology differences on Cu in two different electrolytes, 1 M LiFSI MTBE-toluene and 1 M LiFSI DME, where 1 M LiFSI MTBE-toluene shows higher Coulombic efficiency, adapted with permission from Kim et al.³

Three possible mechanisms may lead to these differences in morphologies seen in our study. First potential mechanism is that LiPS promotes nucleation of Li over growth, which increases the overall surface area and side reactions. Another is the electrolyte decomposition on the current collector prior to Li deposition may affect the morphology. This is a mechanism similar to the one presented above, except that the surface layer is in-situ formed rather than prepared through atomic layer deposition. It is possible that the surface SEI on Cu current collector when LiPS is present is more conducive to the a complete coverage, possibly by altering the electrical conductivity or the lithiophilicity. An alternative mechanism may be surface energy of Li in contact with the electrolyte. It has been suggested that the solvation energy of the electrolyte can dictate the Li surface energy at the Li-electrolyte interface, thereby impacting the morphology of Li.⁴ It could be that the presence of LiPS alters the solvation energy that the high surface area Li morphology is favored.

We have made the following changes to provide these explanations to the readers.

Changes to the manuscript

- Addition of the discussion in the Lithium Metal Anode Cyclability, Morphology and Interphase section: “Similar phenomena have been previously reported, where Li deposited in islands rather than covering the entire surface area have been correlated with improved CE.^{2,3} A similar relationship can be observed for the 1 M LiTFSI DME-TTE 1:1 case, shown in Fig. 4f-g. A potential mechanism is that LiPS promotes nucleation of Li over growth, which leads to the formation of small nuclei, increasing the overall surface area and side reactions. It is possible that electrolyte decomposition layer on the current collector prior to Li deposition may lead to different nucleation behavior. An alternative mechanism may be surface energy of Li in contact with the electrolyte. It has been suggested that the solvation energy of the electrolyte can dictate the Li surface energy at the Li-electrolyte interface, thereby impacting the morphology of Li.⁴ It could be that the presence of LiPS alters the solvation energy that the high surface area Li morphology is favored.”

2. More details on the methods about how to obtain solvation energy should be offered for better understanding.

The authors appreciate the reviewer’s comments. We agree that additional details on the measurement can aid the readers without previous exposure to our work. We have made the following changes to our manuscript.

Changes to the manuscript

- Addition of the description in the Solvation Free Energy Measurement of Electrolytes section: “It employs potentiometry of an electrochemical cell with symmetric Li metal electrodes but asymmetric electrolytes in the two half cells. The half reactions consist of the Li/Li⁺ redox couple. Because the identical electrode terms cancel out, only the Li⁺ terms solvated in two disparate electrolytes remain, and the resulting net reaction is the transfer of Li⁺ between two electrolytes. The change in free energy in this process, which is the difference in the solvation free energies of Li⁺ in different electrolyte environments, gives rise to an electromotive force that can be measured. By holding one electrolyte as a reference electrolyte, it is possible to characterize and compare the solvation free energy of various electrolytes relative to a reference electrolyte. It is important to note that while the standard notion for solvation free energy is defined for the dilute limit, in this work, we consider practically more relevant finite concentrations where the solvation free energy includes the concentration effects. In essence, our measured concentration-dependent solvation free energy is the chemical potential of dissolved ions.”

3. Does the level of the 2nd platform affect the capacity of the battery?

Thank you for your question. In Fig. 2, we provide a correlation between solvation free energy and 2nd plateau voltage, and in Fig. 3, we have provide a correlation between solvation free energy and discharge capacity of the Li-S battery. By combining these two pieces of information, we can draw correlations

between the 2nd plateau voltage and the battery discharge capacity. We find that as the 2nd plateau voltage increases, the battery discharge capacity increases. Although such correlations can be drawn, it does not necessarily mean that higher 2nd plateau voltage causes increases in the discharge capacity. As we explained in the main text, we believe that the 1st discharge capacity is closely linked with the cycling efficiency; as efficiency decreases, the amount of charge lost due to irreversible capacity loss or self-discharge can increase, lowering the capacity. A direct link between the 2nd plateau, such as the reaction chemistry that are responsible for the plateau, to the battery capacity is a highly interesting question that can be explored in future studies.

4. Why choose DME and DME-TTE 1:1 for comparison, instead of DME and 4DME, which can exclude the influence of TTE?

We appreciate the great question. Indeed, selecting 4 M DME would bypass the complication of the TTE diluent. The reason we selected DME-TTE 1:1 was due to a practical problem; to conduct DOSY-NMR experiments, sufficient concentration of Li₂S₆ was needed to create sufficient signal. 0.1 M of Li₂S₆ was the threshold for the lowest concentration, and weakly solvating electrolytes such as 4 M DME or DME-TTE 1:4 could not solvate enough LiPS for accurate measurement. However, we realize that the reasoning behind the selection was unclear in the manuscript. We have made the following changes to improve our manuscript.

Changes to the manuscript

- Addition of the sentence in the Solvation Effect on Li-S Battery Voltage Profile section: “These differences are intertwined with the solvation structure. We compared two electrolytes with different solvation strengths, 0.1 M Li₂S₆ in DME and DME-TTE 1:1, using molecular dynamics and diffusion-ordered spectroscopy-nuclear magnetic resonance (DOSY-NMR). (Supplementary Fig. 4-6). For the weakly solvating solvent formulation, DME-TTE 1:1 was used to ensure sufficient solubility of 0.1 M Li₂S₆ for the DOSY-NMR experiment.”

5. In Fig 3e, there seems to be no pattern between solvent energy and initial capacity, why?

Thank you for the question. Indeed, the trend between solvation energy and initial discharge capacity is not as pronounced as some of the other correlations. However, we believe that there is a positive correlation, as shown in Fig. 3e in the manuscript, which shows that weaker solvation leads to greater initial discharge capacity. This may be caused by improved Coulombic efficiency with weaker solvation. As the Coulombic efficiencies can be quite significant, ranging from around 70% to 96%, it can have a large impact on the discharge capacity as well, explaining the differences in the observed discharge capacity.

Reviewer #3 (Remarks to the Author):

This manuscript (Solvation-Property Relationship of Lithium-Sulfur Battery Electrolytes) uses solvation free energy, measured by potentiometric techniques, as a metric to draw solvation-property relationships of Li-S battery electrolytes. Findings indicate that solvation free energy plays a crucial role in determining the voltage profile of Li-S batteries, as well as the solubility of lithium polysulfides (LiPS), cyclability of Li-S full cell, and the performance of Li metal anode. Weaker solvation free energy results in a lower first plateau voltage, higher second plateau voltage, lower LiPS solubility, and superior performance of the Li-S full cell and Lithium metal anode. But, the novelty is not enough for Nature Communications, and some expressions in the manuscripts are still inadequate or ambiguous. I suggest that it is not suitable for publication on this journal:

We are highly grateful for the time and effort spent by the reviewer to read and evaluate our manuscript. As you mentioned, solvation free energy is a critical parameter in dictating many aspects of the Li-S battery performance, and our methodology of quantitatively characterizing it and drawing solvation-property relationships will be important for providing valuable insights. Regarding the novelty of our work, we think that we may have inadequately communicated the novelty and the impact of our work. We believe that this work provides critical quantitative insights regarding Li-S battery electrolytes that were previously unavailable. The experimental measurement of solvation free energy for Li battery electrolytes, which is the key enabler of this study, was developed recently by our group and was previously not available. The novel results obtained through our technique provide quantitative and convincing evidence to elucidate key solvation-property relationships for Li-S battery electrolytes. First, we find a striking correlation between Li-S battery voltage profile and solvation free energy. It is the first time that clear correlations between plateau voltages and solvation energy have been established, which was enabled by our quantitative characterization of the solvation energy. Moreover, we provide a mechanism based on thermodynamic principles that can explain the phenomena. Secondly, we establish a fundamental relationship between solvation free energy and polysulfide solubility. Indeed, it has been known that strongly solvating electrolytes have increased solubility of polysulfide species¹, but in this study, we extend the horizon of knowledge by establishing quantitative relationships between solvation free energy and solubility and also deducing the solubility constant (K_{sp}). Deducing these fundamental relationships and parameters are critical to better understanding the polysulfide dissolution problem, as well as to designing high-performance electrolytes. Lastly, we reveal the relationship between Coulombic efficiency (CE) of Li metal anodes and solvation free energy of Li-S battery electrolytes, where we find that weaker solvation leads to superior CEs. Overall, we believe that significant and novel quantitative solvation-property relationships are presented in this work, powered by our potentiometric method to characterize solvation free energy. In addition, in regards to the ambiguous expression and communication in our manuscript, we believe that owing to your valuable comments, we have revised our manuscript to a level suitable for publication in *Nature Communications*.

1. It is interesting that the initial discharging capacity rises with weak solvating electrolytes. The shutting effect may play a significant role in increasing the internal resistance of the battery. Besides, the authors claim that the loss of sulfur in the anode side will decrease the initial discharging capacity, which also makes sense. However, a quantification of the effect on resistance and the loss of sulfur in the anode side is lacking to represent a more specific result.

The authors appreciate this excellent comment. Certainly, the internal resistance of the Li-S battery is an important property that is dependent on the electrolyte. Physicochemical properties of the electrolyte, such as viscosity and transport properties, as well as the shuttling effect and LiPS decomposition on the anode can critically affect the cell resistance and performance. With the reviewer's comment as a guideline, we

investigated the internal resistance of the batteries by examining the overpotentials during cycling. Fig. R5 shows the differences in the 2nd plateau voltage between cycling at 0.05C and 0.2C, which can tell us about the overpotentials and the internal resistances of the cell. It shows that weaker solvation correlates with larger overpotentials, which may arise from several different factors.

First important component of cell resistance is ohmic, where the bulk electrolyte resistance is a key component. Two important contributors to the electrolyte bulk resistance are electrolyte viscosity and hydrodynamic radius. The intrinsic viscosity of the electrolyte does not show a clear pattern with solvation energy. For example, 4 M DME that has the weakest solvation energy can have the highest viscosity, where DME-TTE 4:6 has the second weakest solvation energy but low viscosity. However, the hydrodynamic radius impacted by the clustering behavior can also contribute to the phenomenon. Weaker solvation leads to incomplete ion dissociation and greater ion aggregation, as shown in Fig. 2e in the manuscript. This increases the hydrodynamic radius and decreases Li⁺ diffusivity, as shown in Fig. 2f in the manuscript. Therefore, lower ionic conductivity caused by ion clustering behaviors stemming from weak solvation may be an important contributor to the increased overpotentials and cell resistance.

Another important contributor to cell resistance is charge-transfer resistance. It is known that LiPS species can act as redox mediators that can accelerate the redox reaction,¹ and therefore the presence and concentration of LiPS can be an important factor in the charge-transfer resistance. It is possible to see that the LiPS solubility, shown in Fig. 3b in the main text, inversely correlates with overpotential. This hints that charge-transfer resistance may be a significant contributor to the overall cell resistance and that LiPS solubility can be a critical factor, which is also in agreement with previous studies.^{7,8} Lastly, diffusion resistance caused by concentration gradients is an important contributor to cell resistance, particularly at high current densities. This factor is closely related to the electrolyte transport properties which will be impacted by viscosity and aggregation. To summarize, it can be hypothesized that transport properties and charge-transfer kinetics can be affected by LiPS solubility and electrolyte properties, that lead to the correlation of weak solvation to increased overpotentials. However, it is important to note that many factors impact the overall resistance, and it is difficult to attribute the trend to a single factor, which also explains the relatively large spread in this correlation.

Changes to the manuscript

- Addition of Supplementary Fig. 7
- Addition of the paragraph under Polysulfide Solubility and Li-S Battery Cyclability section: “Although weaker solvation is beneficial in improving CE and the initial discharge capacity, it can take a toll on the kinetics. Supplementary Fig. 7 shows that the 2nd plateau overpotential, obtained through the differences between cycling at 0.05C and 0.2C, increases with weaker solvation. This may be a result of insufficient concentration of LiPS that act as redox mediators to accelerate the charge-transfer reaction.^{7,8} The kinetics seems to be play an important role in cycling stability as well, seen by the fact that capacity retention does not increase with weaker solvation, but rather peaks between solvation energies of -10 to -12 kJ mol⁻¹ (Supplementary Fig. 8), a phenomenon that corroborates with recent findings.⁹ In addition, if solvation is further weakened and LiPS solubility is further suppressed, kinetic effects can take over where the discharge capacity decreases with weaker solvation (Supplementary Fig. 1).^{1,10} Therefore, regulating solvation strength is crucial for balancing electrochemical stability and kinetics to achieve optimal battery performance.”

● G4 ● DME ● 2 M DME ● THF ● DOL-DME ● DME-TTE 6:4 ● DME-TTE 1:1 ● DME-TTE 4:6 ● 4 M DME

Fig. R5. Correlation of 2nd plateau overpotential, obtained by the plateau voltage differences in 0.2C and 0.05C, to solvation energy.

2. In the discussion of voltage profiles, the authors represent that electrolytes may display different voltage plateaus and perform a theoretical simulation to explain this. There is no explanation of how voltage plateaus affect battery performance, which can be confusing.

We are grateful for the reviewer's helpful feedback. As you implied, voltage profiles are critical to the battery performance and operation. It is important in two regards. First and foremost, the energy density of a battery depends on the voltage; the energy content of a battery is governed by voltage and capacity. Secondly, from an operational point of view, the voltage profile is critical for the state of charge (SOC) measurement. The thermodynamic voltage profile, or the open circuit voltage (OCV) is an important indicator of the battery's SOC, which will indicate how much charge remains in the battery at any given moment. However, as we show in this work, the thermodynamic voltage profile changes with the electrolyte composition for Li-S batteries. Therefore, relationship between the electrolyte and the voltage profile will be crucial for an accurate measurement of the SOC of the battery. We have added the above explanations into the main text of our manuscript, as below.

Changes to the manuscript

- Revised the following part in the Solvation Effect on Li-S Battery Voltage Profile section: "The voltage of a battery is at the core of a battery. Battery voltage, alongside capacity, dictates the energy content of a battery cell. Battery voltage also impacts the computability and stability of other battery components. From an operational perspective, thermodynamic voltage profile is an

important indicator of the battery's state of charge, which will indicate how much charge remains in the battery at any given moment. Although in most Li-based battery chemistries, the thermodynamic voltage is agnostic to the electrolyte, the Li-S battery is different; it has a unique chemistry with active materials in both solid and solvated phases.”

3. Since the solvation energy is of great significance to the kinetics and thermodynamics of Li-S batteries, can authors provide a solvation energy standard (or value range) for the future optimization of Li-S electrolytes?

We appreciate the great suggestion. From our analyses, we find that weaker solvation is favored for electrochemical stability at the sulfur cathode and lithium metal anode. However, it is unlikely that the weakest solvating electrolyte is practically the best performing electrolyte, primarily due to kinetic limitations. At a certain threshold, kinetic limitations will render the battery practically dysfunctional, as was shown in Supplementary Fig. 1. From our studies, we recommend a solvation free energy range of -12 to -4 kJ mol⁻¹ vs. 1 M LiFSI DEC. We believe that this range provides a reasonable balance between electrochemical stability and reaction kinetics. However, it is important to note that this range of solvation energies is contingent on room temperature operation at moderate current densities ranging from C/10 – C/3. Depending on the operation conditions, solvation energies needed for optimal performance is likely to be different. In addition, in this study we assumed no additives were employed, but in practical electrolytes, additives such as LiNO₃ will be used. This can significantly affect the optimal solvation energy value as well. We have made the following revisions to our manuscript.

Changes to the manuscript

- Revised the conclusion: “In this study, we employed solvation free energy measured through potentiometric techniques as a metric to draw solvation-property relationships of Li-S battery electrolytes. We found that more positive solvation energy (weaker solvation) leads to lower 1st voltage plateau and higher 2nd voltage plateau, which stems from the free energies of LiPS solvated in electrolyte relative to the insoluble Li, S and Li₂S. Weaker solvation also leads to lower LiPS solubility and it was found that solvation free energy is directly proportional to -ln(K_{sp}). Solubility is linked to both Li-S battery CE and initial discharge capacity. We also found that weaker solvation leads to superior Li metal CE and LiPS generally has a negative effect on the CE. Although weaker solvation has proven to be beneficial to full cell and Li metal CE, weaker solvation is detrimental to the internal resistance of the cell, calling for a balance. We believe that these solvation-property relationships can be useful in designing electrolytes for Li-S batteries, and anticipate that solvation free energy can be correlated to other key electrolyte properties and reveal important fundamental insights.”

4. Will the solvation energy be the only factor that influences the CE and initial capacity? Adding fluorinated solvents into the electrolytes can also enhance the CE and lower the battery resistance. For instance, the generation of LiF by the decomposition of F-contained solvents can protect the anode from further corrosion of LiPSs. Discussion of this section is lacking in the manuscript.

The authors greatly appreciate this excellent comment. We would like to first mention that we agree that such a discussion was lacking in the manuscript and thus provided changes to our manuscript. The reviewer is correct that solvation energy is likely not the only factor that affects the CE and the discharge capacity. We completely agree that the molecular details and the chemical pathways of electrolyte components can lead to different results in the battery performance. Therefore, we have added the following discussion.

Changes to the manuscript

- Addition of the discussion in Polysulfide Solubility and Li-S Battery Cyclability section: “With that, it is also important to note that solvation energy is not the sole factor in governing these properties; details in molecular structures and chemical reaction pathways that lead to different passivation layers are important contributors that must be taken into account for a holistic understanding.”

5. The long-lasting effect of solvation energy on the full cell should be discussed. Authors should provide the long cycle performance of Li-S batteries using different electrolytes.

We appreciate this highly constructive feedback. Li-S battery cycling performance, cycled at 0.2C for 150 cycles, is shown below in Fig. R6. It is interesting to see that DOL-DME provides the best capacity retention after 150 cycles, with a relatively large margin ahead of all other electrolytes. Electrolytes that follow DOL-DME are DME-TTE 6:4 and DME-TTE 1:1. It can be seen that overall, electrolytes with moderate solvating power possess the greatest cycling performance. The fact that the full-cell cycling performance does not directly correlate with the Coulombic efficiencies (which increases more or less monotonically with weaker solvation) hints that kinetic factors play an important role—capacity may not be utilized due to limited kinetics. As we saw in our response to the first comment, overpotential increases with weaker solvation. Among the electrolytes with moderate-strong solvation power between -10 to -12 kJ mol⁻¹, DOL-DME stands out with the best cycling performance potentially due to its exceptionally high Li metal Coulombic efficiency, which likely arises from the SEI-forming properties of DOL. As a whole, cycling performance is a result of complex interplay of all of the properties mentioned in our work, including LiPS solubility, electrochemical stability and kinetics. We have made the following changes to the manuscript to reflect this point.

Changes to the manuscript

- Addition of Supplementary Fig. 8.
- Addition of the paragraph under Polysulfide Solubility and Li-S Battery Cyclability section: “Although weaker solvation is beneficial in improving CE and the initial discharge capacity, it can take a toll on the kinetics. Supplementary Fig. 7 shows that the 2nd plateau overpotential, obtained through the differences between cycling at 0.05C and 0.2C, increases with weaker solvation. This may be a result of insufficient concentration of LiPS that act as redox mediators to accelerate the charge-transfer reaction.^{7,8} The kinetics seems to play an important role in cycling stability as well, seen by the fact that capacity retention does not increase with weaker solvation, but rather peaks between solvation energies of -10 to -12 kJ mol⁻¹ (Supplementary Fig. 8), a phenomenon that corroborates with recent findings.⁹ In addition, if solvation is further weakened and LiPS solubility is further suppressed, kinetic effects can take over where the discharge capacity decreases with weaker solvation (Supplementary Fig. 1).^{1,10} Therefore, regulating solvation strength is crucial for balancing electrochemical stability and kinetics to achieve optimal battery performance.”

Fig. R6. Li-S battery cycling performance of different electrolytes.

Reviewer #4 (Remarks to the Author):

Li-S batteries have the potential to replace lithium-ion batteries once the major technical challenges arising from polysulfide transport and the redox mechanism are understood. However, unlike the conventional transition metal oxide chemistry in Li-ion batteries, where the active materials remain solid throughout the redox process, the dissolution of lithium polysulfide species (the intermediate redox species of the S/S²⁻ redox couple) into the electrolyte poses a significant challenge. This is especially true when the average cell potential at different processes and cycling performance heavily depends on the solvation structure of lithium polysulfide in the electrolyte. Understanding this solvation-property-performance relationship is a challenging and crucial problem that requires careful attention.

The work presented in manuscript NCOMMS-23-26016 utilizes a relatively simple potentiometric measurement method that the authors recently reported. It aims to draw a correlation between the solvation free energy of the electrolyte and its impact on the solvated lithium polysulfide. The concept and experimental method presented in the manuscript are novel and should be reproducible. I agree that the work is significant enough to further advance Li-S technology. However, there are several specific questions I would like the authors to address (see below). In conclusion, I believe the work is worthy of publication in Nat. Commun. once the following comments have been adequately addressed.

The authors are highly grateful for your appreciation of our work. Indeed, practical Li-S batteries hold the potential to revolutionize battery technology and enable electrification of many applications. We agree that the results presented in this work can deepen our understanding of the Li-S battery chemistry and also help advance the Li-S battery technology. We are also appreciative of your constructive feedback. We have provided a point-by-point response to your comments and questions below.

1. The author's statement regarding the solvating energy and its impact on lithium polysulfide is based on a limited number of electrolyte systems, most of which are considered moderately solvating. However, it is well documented by literature review that polysulfide speciation and chemistry can be significantly altered by different classes of electrolyte systems (Joule, 2021, 5, 2323). Considering the importance of understanding the effects of different electrolyte systems, it would be valuable for the authors to comment on the potential impact of alternative solvents, including dimethylsulfoxide (DMSO) and dimethylacetamide (DMA), which are classic examples. By studying the effects of such systems, the authors could provide a more comprehensive understanding of how different solvents influence the solvation energy of lithium polysulfides. This would help avoid overgeneralizing the concept and provide a more nuanced perspective on the solvation-property-performance relationship in Li-S batteries.

Thank you for an excellent comment. As you mentioned, DMSO and DMA are strongly solvating solvents also commonly deployed for Li-S battery electrolytes. These strongly solvating solvents with highly negative solvation free energies¹¹ are expected to show properties that are consistent to our correlations. The voltage profile is expected to have a high first plateau and a low second plateau. As an example, we conducted an experiment to examine the voltage profile of 1 M LiTFSI in DMSO.

Fig. R7. Voltage profiles of 1 M LiTFSI in DME, DME-TTE 1:1 and DMSO.

It can be seen in Fig. R7 that DMSO has a high 1st plateau and a low 2nd plateau, consistent with our correlation and expectations. This result is also consistent with previous observations.^{12,13} It is, however, important to note that DMSO shows significant sloping behavior compared to DME and DME-TTE. This may be related to the high solvating power of DMSO leading to different speciation of LiPS, including the S₃⁻ radical.¹³ These differences in chemistry that arise from highly disparate solvating strengths can make direct extrapolation difficult. In addition, measuring and directly comparing the LiPS solubility also becomes challenging, because the UV-Vis absorption spectrum will shift due to other species including S₃⁻ radical. Therefore, direct comparison of the spectra to characterize concentration will be difficult. For these reasons, we kept our discussion around moderate to weakly solvating electrolytes. However, as you suggested, we recognize that it would be beneficial to include the impact of strongly solvating solvents such as DMSO and DMA, and we have revised our manuscript as below.

Changes to the manuscript

- Addition of the sentence in the Solvation Effect on Li-S Battery Voltage Profile section: “This correlation can be extended to strongly solvating solvents such as dimethyl sulfoxide (DMSO) (Supplementary Fig. 3).”

2. In the main text, the author acknowledges that most Li-S batteries often employ an excessive amount of Li anode, which poses challenges in accurately assessing the anode's health through Coulombic efficiency (CE). This acknowledgment is indeed correct. Therefore, it would be highly beneficial for the authors to provide the N/P ratio. While the Li metal thickness is provided in the experimental method, without knowing the area of the Li anode, it renders great difficulty for readers to assess the practical relevance and potential implications of the authors' findings in Fig. 4

The authors appreciate the reviewer's suggestion. In this study, we employed 500 μm thick Li with an area of 1 cm^2 . The electrodes were cut into disks with mass loading of 1-2 mg cm^{-2} . With these values, the N/P ratio ranges are 29.9-59.8. We have revised the manuscript to include this information.

Changes to the manuscript

- Addition of "The N/P ratio ranges are 29.9-59.8" in the methods section.

3. In the introduction, the authors claim that Li-S batteries are the potential candidates battery technology for eVTOL applications. However, it is worth noting that eVTOL technology requires high volumetric density batteries that can deliver high power (i.e., high C-rate) during take-off and landing. Unfortunately, these are precisely the areas where Li-S battery technology currently faces challenges and limitations. To address this concern, it would be beneficial for the author to provide a more detailed comment explaining why they believe Li-S battery technology is still considered adventurous despite its current weaknesses in high-power applications. They could discuss potential strategies, advancements, or ongoing research efforts aimed at improving the power capabilities of Li-S batteries for demanding applications like eVTOL. This would provide a clearer understanding of the potential viability of Li-S batteries in the context of eVTOL technology.

This is an excellent point. We completely agree that electric aviation, including eVTOLs, calls for high power and high volumetric energy density in addition to high specific energy density. And these demanding requirements make electric aviation so challenging. Among these requirements, specific energy density poses the most stringent restrictions,¹⁴ and is the criteria that rules out many battery chemistries. The currently commercialized Li-ion battery chemistries have cell-level energy densities in the 200-300 Wh kg^{-1} , and even the theoretical energy densities of these battery chemistries are far from the required values for aviation. The high theoretical energy density of the Li-S battery is the primary reason attribute that makes it attractive as a next-generation chemistry for electric aviation. Now, as the reviewer astutely mentioned, delivering high power is an important challenge for Li-S batteries. However, we believe there are pathways to improve kinetics. For example, the use of catalysts¹⁵, electrode engineering^{16,17} and temperature elevation,⁸ can aid with improving kinetics and delivering high power. With these improvements, Li-S batteries may be a potential solution for electric aviation. In fact, companies such as Lyten and Theion have plans to deploy Li-S batteries for eVTOL and aviation applications. It is also important to note that lithium-ion chemistries, despite the challenges in simultaneously achieving high specific energy density and high power, are also considered for commercial eVTOLs for urban aviation. Because these applications have short range requirements, typically up to about 100 miles, it has been demonstrated that LIBs can be used for eVTOLs. However, for longer-range electric aviation, high energy density becomes more critical, and Li-S would be more promising. Overall, although Li-S batteries can be potential solutions for eVTOLs in the future, it may be more reasonable to focus less on eVTOLs and more on electric aviation as a whole. We recognize that our wording in the manuscript can cause confusion and would like to change the introduction in the following manner:

Changes to the manuscript

- Revised the sentence “With the rapidly decreasing battery costs and growing concerns for climate change, electric vehicles are quickly becoming the future of the passenger vehicle market.¹ However, applications such as long-haul trucking and aviation remain difficult to electrify, and batteries with much higher energy densities are in demand.³” in the first paragraph.

4. In the experimental method, can the author clarify was LiNO_3 used as electrolyte additive?

Thank you for the suggestion. LiNO_3 was absent in all electrolytes. We have made the changes below.

Changes to the manuscript

- Addition of the sentence “In all electrolytes, LiNO_3 was not added as an additive.” in methods section.

5. I understand the authors colour-coded their figure (example include Fig. S3, 5, and 6.) that is consistent with the types of electrolytes used (first mentioned in Fig. 2). However, re-labeling the legend in these figure can provide clarity and avoid potential confusion for reader.

Thank you very much for a great comment. We agree that it would be much better to provide legends in these figures to help the reader understand and match the color to the electrolyte. We have revised our Supplementary Figures to include the legends in each of the figures. Please find the revised figures below.

● G4 ● DME ● 2 M DME ● THF ● DOL-DME ● DME-TTE 6:4 ● DME-TTE 1:1 ● DME-TTE 4:6 ● 4 M DME

Fig. R8. Correlation between solvation energy and the average discharge voltage, showing a weak positive correlation.

● DME
 ● 2 M DME
 ● THF
 ● DOL-DME
 ● DME-TTE 6:4
 ● DME-TTE 1:1
 ● DME-TTE 4:6
 ● 4 M DME

Fig. R9. The relationship between $-\ln(K_{sp})$ and solvation energy for three different possible dissolution reactions, showing the best fit for the second reaction in b.

● G4 ● DME ● 2 M DME ● THF ● DOL-DME ● DME-TTE 6:4 ● DME-TTE 1:1 ● DME-TTE 4:6 ● 4 M DME

Fig. R10. Correlation between solvation energy and the 1st and 2nd plateau capacities, showing an increase with weaker solvation.

Changes to the manuscript

- Added legends to Supplementary Figs. 3, 5, 6

References

1. Liu, Y. *et al.* Electrolyte solutions design for lithium-sulfur batteries. *Joule* **5**, 2323–2364 (2021).
2. Oyakhire, S. T. *et al.* Electrical resistance of the current collector controls lithium morphology. *Nat. Commun.* **13**, 3986 (2022).
3. Kim, S. C. *et al.* Data-driven electrolyte design for lithium metal anodes. *Proc. Natl. Acad. Sci.* **120**, e2214357120 (2023).
4. Boyle, D. T. *et al.* Correlating Kinetics to Cyclability Reveals Thermodynamic Origin of Lithium Anode Morphology in Liquid Electrolytes. *J. Am. Chem. Soc.* **144**, 20717–20725 (2022).
5. Holoubek, J. *et al.* Electrolyte design implications of ion-pairing in low-temperature Li metal batteries. *Energy Environ. Sci.* **15**, 1647–1658 (2022).
6. Perez Beltran, S., Cao, X., Zhang, J. G. & Balbuena, P. B. Localized High Concentration Electrolytes for High Voltage Lithium-Metal Batteries: Correlation between the Electrolyte Composition and Its Reductive/Oxidative Stability. *Chem. Mater.* **32**, 5973–5984 (2020).
7. Zhang, X. Q. *et al.* Electrolyte Structure of Lithium Polysulfides with Anti-Reductive Solvent Shells for Practical Lithium–Sulfur Batteries. *Angew. Chemie - Int. Ed.* **60**, 15503–15509 (2021).
8. Gao, X. *et al.* Electrolytes with moderate lithium polysulfide solubility for high-performance long-calendar-life lithium–sulfur batteries. *Proc. Natl. Acad. Sci.* **120**, e2301260120 (2023).
9. Li, Z. *et al.* Correlating Polysulfide Solvation Structure with Electrode Kinetics towards Long-Cycling Lithium–Sulfur Batteries. *Angew. Chemie Int. Ed.* **n/a**, e202309968 (2023).
10. Zhang, S. S. Liquid electrolyte lithium/sulfur battery: Fundamental chemistry, problems, and solutions. *J. Power Sources* **231**, 153–162 (2013).
11. Marcus, Y. *Ions in Solution and their Solvation. Ions in Solution and their Solvation* (2015). doi:10.1002/9781118892336.
12. Yu, X. & Manthiram, A. A class of polysulfide catholytes for lithium-sulfur batteries: Energy density, cyclability, and voltage enhancement. *Phys. Chem. Chem. Phys.* **17**, 2127–2136 (2015).
13. He, Q., Gorlin, Y., Patel, M. U. M., Gasteiger, H. A. & Lu, Y.-C. Unraveling the Correlation between Solvent Properties and Sulfur Redox Behavior in Lithium-Sulfur Batteries. *J. Electrochem. Soc.* **165**, A4027–A4033 (2018).
14. Viswanathan, V. *et al.* The challenges and opportunities of battery-powered flight. *Nature* **601**, 519–525 (2022).
15. Zhou, J. *et al.* Deciphering the Modulation Essence of p Bands in Co-Based Compounds on Li-S Chemistry. *Joule* **2**, 2681–2693 (2018).
16. Moon, S. *et al.* Encapsulated Monoclinic Sulfur for Stable Cycling of Li–S Rechargeable Batteries.

Adv. Mater. **25**, 6547–6553 (2013).

17. He, J. & Manthiram, A. Long-Life, High-Rate Lithium–Sulfur Cells with a Carbon-Free VN Host as an Efficient Polysulfide Adsorbent and Lithium Dendrite Inhibitor. *Adv. Energy Mater.* **10**, 1903241 (2020).

REVIEWERS' COMMENTS

Reviewer #2 (Remarks to the Author):

The authors have revised our comments with appropriate rigor. Here we think this revision can be published in Nature Communications.

Reviewer #3 (Remarks to the Author):

The author has made impressive efforts in addressing the issues, but I still have some questions regarding the relationship between the voltage platform and battery performance (Question 2), as well as the factors influencing the CE of the lithium metal anode (Question 4). Especially after reading the recent research paper titled "Electrode potential influences the reversibility of lithium-metal anodes" published in Nature Energy (DOI: <https://doi.org/10.1038/s41560-022-01144-0>), I have some confusion about the explanations provided by the author. In this paper, it is mentioned that the electrode potential of metallic lithium is influenced by its coordination state, and a higher thermodynamic potential of lithium is advantageous for enhancing CE. Under this premise, the discussion on electrode potential in the article should consider not only the cathode side but also the influence of solvation on the potential of the anode side. The lithium metal cannot be simply regarded as a reference electrode anymore. I believe that incorporating a theoretical study simultaneously considering the potentials of both the cathode and anode will make this article more innovative and scientifically rigorous.

Response Letter to Reviewers for “Solvation-Property Relationship of Lithium-Sulfur Battery Electrolytes”

We are grateful to the reviewers for providing valuable comments and feedback to improve our manuscript. We believe that the comments and questions raised by the reviewers have helped us clarify our arguments and show the novelty of our work. In the letter below, we have responded to each question raised by the reviewers in blue. The revisions made to the main text of our manuscript have been highlighted to aid the reviewers.

Reviewer #2 (Remarks to the Author):

The authors have revised our comments with appropriate rigor. Here we think this revision can be published in Nature Communications.

The authors are grateful for the reviewer’s acknowledgement, and the constructive feedback that has helped improve the robustness and clarity of our manuscript.

Reviewer #3 (Remarks to the Author):

The author has made impressive efforts in addressing the issues, but I still have some questions regarding the relationship between the voltage platform and battery performance (Question 2), as well as the factors influencing the CE of the lithium metal anode (Question 4). Especially after reading the recent research paper titled "Electrode potential influences the reversibility of lithium-metal anodes" published in Nature Energy (DOI: <https://doi.org/10.1038/s41560-022-01144-0>), I have some confusion about the explanations provided by the author. In this paper, it is mentioned that the electrode potential of metallic lithium is influenced by its coordination state, and a higher thermodynamic potential of lithium is advantageous for enhancing CE. Under this premise, the discussion on electrode potential in the article should consider not only the cathode side but also the influence of solvation on the potential of the anode side. The lithium metal cannot be simply regarded as a reference electrode anymore. I believe that incorporating a theoretical study simultaneously considering the potentials of both the cathode and anode will make this article more innovative and scientifically rigorous.

The authors would like to thank the reviewer for the important question. First of all, I would like to establish that, as the reviewer correctly mentioned, the electrode potential will play an important role at the Li metal anode side. In Fig. 4, we observe that weakly solvating electrolytes will lead to superior CEs for the Li metal anode, a finding that is consistent with our initial findings in our earlier work.¹ Because solvation energy will alter the Li⁺/Li electrode potential, a finding outlined in the Nature Energy paper referenced by the reviewer, it can be said that solvation energy, electrode potential and CE are all correlated.

Now, to address the reviewer’s question: “the discussion on electrode potential in the article should consider not only the cathode side but also the influence of solvation on the potential of the anode side.” We believe that our analysis in Fig. 2 already includes the effect of Li⁺/Li electrode potential shifting with electrolytes due to solvation energy. Let us consider two different electrolytes, EL1 and EL2, with EL1 having stronger solvation (Fig. R1). Li⁺/Li redox potential is higher for EL2 by ΔE . Now, let us consider LiCoO₂ as a hypothetical cathode material to be paired with Li metal anode, with the reduction half reaction of $\text{Li}^+ + \text{CoO}_2 + \text{e}^- \rightarrow \text{LiCoO}_2$. The half reaction involving EL2 is still higher in potential than EL1 by the same

difference ΔE as in the Li^+/Li half reaction. This is because the Li^+ is the only solvated phase in both half reactions. Therefore, when you couple the two half reactions (for example, Li metal and LiCoO_2 interfacing with EL1), the full cell voltage involving the two electrolytes are exactly the same E_1 , because the solvated Li^+ terms cancel out. This is the reason why the electrolyte does not alter the equilibrium voltages of many lithium-metal and lithium-ion batteries. Now, turning to Li-S battery case, let us imagine the reaction $\text{Li}^+ + \text{Li}_2\text{S}_x + e^- \rightarrow \text{Li}_2\text{S}$. Again, the Li^+ terms are canceled out, but in this case the two electrolytes have different potentials because Li_2S_x terms are also solvated in different electrolytes. Therefore, the two reactions involving LiPS species are offset by a voltage difference $\neq \Delta E$. The resulting net reactions by coupling with the Li^+/Li half reaction have different cell voltages. In summary, we are already accounting for the fact that Li electrode potential moves with electrolytes, but because the potential on the cathode side moves in the same direction and amount, the Li^+ effect is canceled out, and we are probing the effects of the dissolved LiPS species.

Fig. R1. The electrode potential dependence on electrolytes.

References

1. Kim, S. C. *et al.* Potentiometric Measurement to Probe Solvation Energy and Its Correlation to Lithium Battery Cyclability. *J. Am. Chem. Soc.* **143**, 10301–10308 (2021).